# THIR: Topological Histopathological Image Retrieval

Anonymous Full Paper
Submission 8

## 001 Abstract

According to the World Health Organization, breast cancer claimed the lives of approximately 685,000 women in 2020. Early diagnosis and accurate clinical decision-making are critical in reducing this global burden. In this study, we propose THIR, a novel Content-Based Medical Image Retrieval (CBMIR) framework that leverages topological data analysis—specifically, Betti numbers derived from persistent homology—to characterize and retrieve histopathological images based on their intrinsic structural patterns. Unlike conventional deep learning approaches that rely on extensive training, annotated datasets, and powerful GPU resources, THIR operates entirely without supervision. It extracts topological fingerprints directly from RGB histopathological images using cubical persistence, encoding the evolution of loops as compact, interpretable feature vectors. The similarity retrieval is then performed by computing the distances between these topological descriptors, efficiently returning the top-$K$ most relevant matches.

Extensive experiments on the BreaKHis dataset demonstrate that THIR outperforms state-of-the-art supervised and unsupervised methods. It processes the entire dataset in under 20 minutes on a standard CPU, offering a fast, scalable, and training-free solution for clinical image retrieval.

## 1 Introduction

Cancer is a leading cause of death worldwide, with nearly 10 million deaths in 2020, or nearly one in six deaths [1]. Breast cancer accounts for 25% of all cancers in women worldwide, and about 685,000 women lost their lives due to breast cancer in 2020 [2]. Histopathology is the gold standard for cancer diagnosis [3], which involves extracting tissue specimens from suspicious areas to prepare a glass slide for a microscopic examination [4]. However, this examination might have some human errors or intraobserver variability. Singh, et al. [5] made an extensive review of the errors in cancer diagnosis. Accurate cancer diagnosis and grading rely on many factors, including the knowledge, experience, and skills of pathologists [6], which can increase the rate of human error in diagnosis. Diagnosis is a high-risk area of errors, including missed, inappropriately delayed, or wrong diagnoses [6]. Digital pathology, by employing Deep Learning (DL) and Machine Learning (ML) techniques, has a significant impact on decreasing human errors by providing a second opinion for pathologists [7]. An image retrieval tool that finds cases with similar morphological features can help diagnose rare diseases and unusual conditions that may not have enough cases available to develop accurate supervised classification models [8].

While Content-Based Image Retrieval (CBIR) has been under investigation for decades [9], only with the emergence of digital pathology and DL, the studies have begun to focus on image search and analysis in histopathology [10]. Content-Based Medical Image Retrieval (CBMIR) offers a new approach to computational pathology [11]. CBMIR provides the top $K$ with patches similar to the query from the previously diagnosed and treated cases. This can assist pathologists in tackling the above-mentioned errors. The main base of CBMIR is similarity measurement, which considers features including texture, shape, intensity, etc., and compares them with the previous cases [12]. This retrieval helps pathologists receive not only the labels, but also patches similar to their query. This can increase the explainability of the methods since pathologists can analyze the texture and compare it with similar cases based on their expertise.

Visual examination of the patterns of the tissue in a monitor is a task usually relegated to pathologists or computational biologists. CBMIR can assist pathologists in analyzing and managing a large volume of images to enhance diagnosis, collaboration, education, research, and the decision-making processes [13]. By searching and retrieving visually similar images with their labels, pathologists have more information to detect the abnormalities. CBMIR not only increases the accuracy of cancer diagnosis but also speeds up the process of consulting peers. It can support remote consultation and collaboration between pathologists from all over the world [14]. In addition, CBMIR is a crucial tool in research for exploring large histopathological image archives to discover and identify new patterns, trends, graphs, and correlations between cancer grades [13]. For example, it can be a teaching tool in histopathological education, allowing trainers to study and compare various histopathological patterns.

CBMIR consists of ranking images concerning a query image based on visual similarities with a typical workflow, as illustrated in Figure 1. Following [15], it has two phases: offline and online.

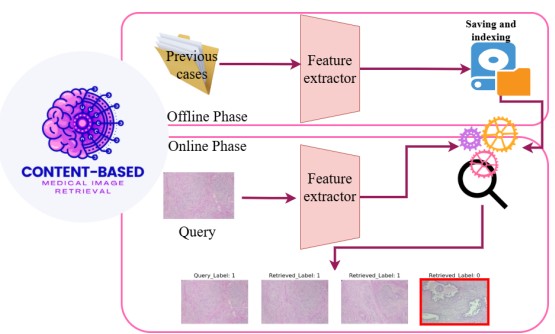

**Figure 1.** shows the main workflow of a CBMIR. It contains two main phases, offline and online. The same FE applies in both phases to extract features. Then, the Euclidean distance as a distance measurement function is applied to find the top-3 similar patches. On top of the query image and the retrieved images, their labels are mentioned.

The offline phase includes extracting features of the previous cases and indexing them. In the online phase, a query image, which is an unseen image for the Feature Extractor (FE), is fed to the same FE as in the offline phase to extract its features. To identify those most relevant images, a similarity function is applied between the extracted features of the query and the previous cases. Then, the top-$K$ most relevant images from the previous cases are retrieved. This is similar to the traditional workflow in hospitals using Atlas books [16].

DL-based methods, particularly Convolutional Neural Networks (CNNs), have shown great success in CBMIR tasks by automatically learning hierarchical feature representations. However, these models often require large amounts of labeled data, are prone to overfitting, suffer from limited GPU resources, and lack interpretability. In this paper, THIR focuses on the topological information of images and texture features extracted from Topological Data Analysis (TDA). TDA provides hand-crafted, mathematically interpretable features, such as Betti values, that describe the global structure of images. While DL focuses on local patterns and appearance, TDA emphasizes global connectivity and shape, offering complementary information [17].

To the best of the authors' knowledge, this is the first study to apply TDA to CBMIR in digital pathology. The main contributions of this paper are as follows:

- **THIR** is a fully unsupervised method that eliminates the need for labeled datasets, addressing one of the key challenges in medical image analysis.

- Through cubical persistence, we capture unique topological signatures by tracking how topological structures develop across the color channels of the images.

- **THIR** is computationally efficient; the entire feature extraction process for the training set using THIR takes approximately 20 minutes, resulting in a compact and informative topological feature vector.

- Unlike ML-/DL-based methods, **THIR** does not require hyperparameter tuning, GPU resources, and a training phase, making it simple, scalable, and accessible.

- The accuracy of **THIR** surpasses that of recently published supervised and unsupervised methods in CBMIR tasks.

## 2   Related work

The performance of CBMIR mainly depends on the choice and performance of the FE method. A high-quality FE algorithm can improve the precision of the search engine [18]. There are some common descriptors, such as Scale-invariant feature transform (SIFT) [19], Local Binary Patterns (LBP) [20], Histogram of Oriented Gradient (HOG) [21], edge histogram descriptor [22], and Gabor filter [23] for texture analysis, which explore the local features of the images. Local features refer to the general pattern of images, such as a point, edge, or small image patch.

Artificial intelligence (AI) enabled CBMIR for an effective diagnosis in [24]. Then, as CNNs show their power and high effectiveness in extracting features, they have become increasingly used for CBMIR. DL-based models yield high-performance search engines by extracting features of images for tasks related to CBMIR. Many recent studies [25–28] were dedicated to exploring the performance of different DL-based methods in CBMIR.

Among the various types of DL-based methods, Auto Encoders (AEs), GANs, and Siamese networks [29] have a special place as FE in the CBMIR task. In [14], the author reported 9.33 and 6.59 hours of training time for training a Convolutional Auto Encoder (CAE) and Federated Learning (FL) CAE, respectively. [14] claims that FedCBMIR is faster compared to traditional CBMIR using the same CAE structure and the same GPU type (NVIDIA GeForce RTX 3090). FedCBMIR provides 98% accuracy, and the UCBMIR in [30] yields 93% accuracy at the top-5 on the BreaKHis data set. The Siamese network in [31] obtains 94% an F1-score at the top-5 retrievals for breast cancer. Authors in [32] in Google AI Healthcare proposed an automatic high-level feature extraction on prostate cancer. The obtained results were reported at the top-5 similar patches with an accuracy of 73%. Yottixel [33] uses the DenseNet structure, which is trained on the ImageNet data set for extracting patches without being trained specifically for the CBMIR task.

RetCCL [34] proposes a method based on clustering feature vectors of the patches. In this work, a ResNet50 was trained using contrastive learning. In [35], a Graph Neural Network (GNN) encodes Region of Interest (ROI) graphs into representations using a contrastive loss function in a self-supervised manner. The study in [36] proposes size-scalable CBMIR from databases that consist of whole-slide images (WSIs). This method has addressed scalable retrieval frameworks tailored to WSIs, focusing on efficient indexing and patch-level comparisons to manage the immense size and complexity of histopathological data.

The authors in [37] provide an overview of the TDA methods in biomedicine. After reviewing the recently published literature, this study aims to explore the potential of TDA with CBMIR applications. To the best of our knowledge, this combination has never been explored for CBMIR tasks in breast cancer.

# 3 Material and Methodology

Our methodology consists of two steps. First, we extract the topological feature vectors from the data set and the query. Then, we apply a similarity measurement function to these vectors to find the top-$K$ similar patches to the query from the data set. This provides a search engine based on the images' topological information, resulting in a fast and interpretable model called THIR. The implementation in this study focuses on breast cancer. Figure 2 and Figure 3 illustrate the whole workflow of the proposed methodology.

## 3.1 Data set

BreaKHis data set [38] was created in the PD laboratory in Prana, Brazil, and consists of 7909 microscopic images of breast cancer. This collection contains four different magnifications ($40\times$, $100\times$, $200\times$, and $400\times$)[1] as shown in Table 1. In this binary data set, tissues were stained with Hematoxylin and Eosin (H&E), which is the most common color in histopathological images [39]. Following the previous studies [14, 30, 31] to be able to have comparable results, we resized the images into $240\times240\times3$. Since the images are from the cancerous tissue, they are not affected by image transforming, inverting, zooming in, or rotation by 90 degrees [40].

## 3.2 Topological Data Analysis (TDA)

Topology studies properties of spaces that are invariant under any continuous deformation. Over the past two decades, TDA has proven highly effective in identifying topological structures within images [41].

**Table 1.** The distribution of BreakHis data set.

| Magnification | Benign | Malignant | Total |
|---|---|---|---|
| $40\times$ | 625 | 1370 | 1995 |
| $100\times$ | 644 | 1437 | 2081 |
| $200\times$ | 623 | 1390 | 2013 |
| $400\times$ | 588 | 1232 | 1820 |
| Total | 2480 | 5429 | 7909 |

In the context of histopathological image analysis, TDA has demonstrated strong potential in cancer detection [42]. TDA extracts meaningful patterns by analyzing the homological features of images. These features can quantify the complex topological shapes and geometric structures in the data. The advantage of TDA is that it can effectively process complex and high-dimensional data, capture the global topological structure of the data, and provide a deep understanding of the shape of the data [43]. This means that TDA offers a mathematically robust and interpretable way to capture topological features, especially in medical images where texture, shape, and structure matter.

In this study, we utilize TDA to extract topological features from medical images, specifically focusing on Persistent Homology (PH), one of TDA's most prominent tools. PH quantifies how topological features—such as connected components, loops, and voids—evolve across different scales, represented by a filtration. A filtration is a sequence of nested spaces generated by progressively thresholding the data. Within this framework, a feature *is born* at the threshold where it first appears and *dies* at the threshold where it is merged or disappears. Long-lived features (those with large persistence) are considered topologically meaningful, while short-lived ones are often attributed to noise [44]. A comprehensive overview of TDA and PH is provided in [45].

There are two widely used algorithms for computing homology: simplicial and cubical homology. While simplicial complexes offer greater generality, cubical complexes are more computationally efficient and well-aligned with image data, which naturally reside on regular pixel grids [46, 47]. In the cubical setting, images are modeled as 2D grids of square cells (pixels), and a topological structure is built by thresholding pixel intensities. The resulting binary images are interpreted using 0D (points), 1D (edges), and 2D (squares) elements [48].

As the threshold increases, new topological features appear and disappear, which can be quantified using Betti numbers: $\beta_0$ (number of connected components), $\beta_1$ (number of loops), and $\beta_2$ (number of voids). These values provide compact, interpretable descriptors of image structure and texture. In medical imaging, they serve as powerful shape-based features that are invariant to rotation and intensity changes and are equivalent to spatial scaling, offering a complementary perspective to pixel-based

---

[1]https://www.kaggle.com/datasets/ambarish/breakhis

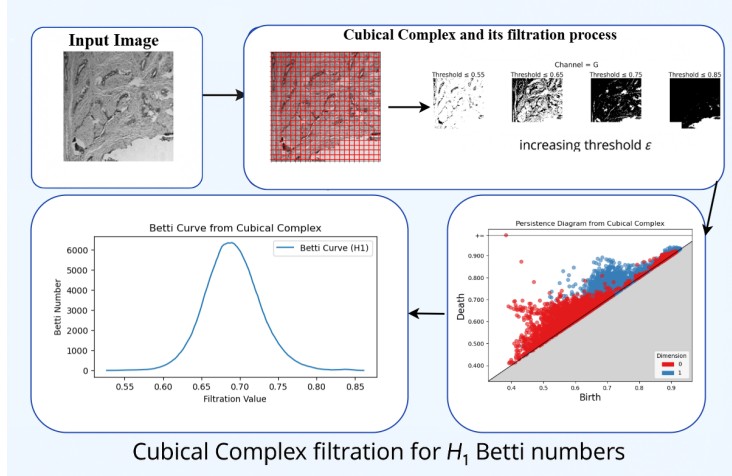

**Figure 2.** THIR model. We first generate persistence diagrams for any input images, utilizing the cubical complex on each channel of the images. Next, we derive our topological feature vectors, represented as the Betti curves. The values of this curve are then input into the CBMIR workflow to produce the results of the search engine.

or deep learning methods. This interpretability is particularly valuable in clinical contexts, where explainability is essential.

Figure 2 illustrates the cubical complex process. The input image is first overlaid with a grid, and each unique intensity value is treated as a threshold value $(th)$. For 8-bit grayscale images, the filtration spans $K_r r \in \mathcal{T} = K_0, \ldots, K_{255}$. As the $th$ value increases, connected components and loops emerge and then merge or vanish. The persistence diagram in the figure summarizes this process: $\beta_0$ features are shown in red and $\beta_1$ in blue. The diagonal line represents $birth = death$, and points far from the diagonal indicate more persistent, and thus more meaningful features [49].

In this study, we focus on $\beta_1$. So, the Betti values are the number of loops in the image. As can be seen in Figure 2, the Betti curve for this data set resembles a bell shape, peaking where most loops are present. Initially, at low thresholds, few features appear due to limited pixel activation. As the threshold increases, more features emerge until a saturation point is reached. At high thresholds, the image becomes nearly black, and topological features disappear. This dynamic is further illustrated in Figure 3, which shows the effect of different threshold values on the R, G, and B channels of an image. The transformation of pixel intensities across channels explains the bell-shaped nature of the Betti curve. It is noteworthy to mention that the RGB color space in the dataset is the default mode, which is aligned with the staining of the tissues. [50] provides a comprehensive overview of different color spaces in digital pathology.

Since our dataset consists of RGB images, we applied the described cubical complex pipeline separately to each channel. Consequently, each image yields three Betti curves—one for each channel.

These topological descriptors are then concatenated together and forwarded to the CBMIR pipeline for downstream analysis and similarity-based image search.

Betti curves are naturally calculated on a non-uniform filtration scale. For instance, let us assume we have $n$ loops represented as $\beta_1 = [(b_1, d_1), (b_2, d_2), \ldots, (b_n, d_n)]$. For homogeneous treatment across datasets, we uniformly select $R = i$ filtration points within the range between the minimum birth value and the maximum death value, denoted as $X = [R_1, R_2, \ldots, R_i]$. A loop $\beta_1(n)$ is considered alive at a filtration point $X_i$ if $b_n \leq X_i \leq d_n$. Thus, we introduce a Betti curve *resolution* $(R)$, which determines the granularity at which topological features are detected such that higher resolutions preserve fine-grained structures but may capture noisy artifacts, while lower resolutions emphasize large-scale features at the cost of missing subtle patterns. The appropriate resolution for a given image set is a modeling choice that balances sensitivity and robustness in tissue analysis.

## 4 Experiments

In this paper, we focus on the application of TDA in digital pathology and analyze the topological patterns of the histopathological images. To do so, our method of choice is the Betti curves of the cubical complexes. The cubic complex from the Gudhi Library[2] works with grayscale images. However, in digital pathology and working with WSIs, color plays a significant role that cannot be discarded from the experiments [40]. To address this, we apply cubical complex persistence separately to each RGB channel, treating them as independent grayscale images.

---

[2]https://gudhi.inria.fr/cubicalcomplex/

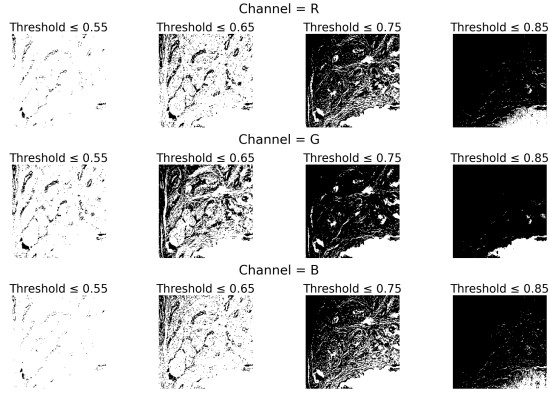

**Figure 3.** shows three channels of an RGB image under different *th* values. These values were defined as a sublevel filtration in [0-1] for the normalized images. Each channel goes through the same *th* values to illustrate the *th* impacts on channels.

The resulting Betti values from each channel are then concatenated to form a comprehensive topological descriptor for each image. With Betti curves sampled uniformly in $R$ resolution, for each image we have $3 \times R$ features. As mentioned above, finding the optimum value for the resolution is challenging and has a direct impact on the final results. As the resolution $R$ increases, more fine-grained topological details are captured, though at the cost of longer computation times. In the case of BreaKHis data set, $R = 200$ offers a strong balance between performance and efficiency, yielding the best accuracy with a much lower computational burden, followed by [45]. So, for the following experiments in this paper, we considered $R = 200$. Therefore, a descriptor representing each image is constructed from the features derived from $\beta_1$ for the CBMIR framework. In the next step, the extracted features for the entire dataset are saved. Subsequently, we compute the features of the test set, which was previously separated from the training set. With 600 features extracted per image, we proceed to the comparison phase. Specifically, we use the Euclidean distance metric to identify the top-$K$ most similar patches from the training set for each query image. After ranking the images based on their smallest distances, the top matching patches are retrieved and visualized along with their corresponding labels to assist pathologists in analysis.

# 5 Results and discussion

When evaluating the performance of CBMIR methods, several important aspects must be considered, including training time, accuracy, and ease of training. In this section, we compare the performance of THIR with other methods across all these dimensions.

## 5.1 Accuracy Comparison

One challenge in this study is the scarcity of comparable methods evaluated on the same dataset under identical conditions (i.e., same $K$ value and magnification). To address this, we provide Table 2 and some more figures, such as Figure 4, to provide a comprehensive overview of the results in this study.

summarizing recent studies on BreaKHis, indicating their magnification and the $K$ value considered. This enables clearer benchmarking of THIR across multiple settings. The evaluation is performed at multiple values of $K$, focusing on key performance metrics: accuracy, recall, precision, and F1-score.

In published studies on CBMIR, the value of $K$ varies across works. Additionally, different magnifications of the BreaKHis dataset have been considered, and only a few studies have applied their methods to all magnifications (40×, 100×, 200×, and 400×). These variations make direct comparison challenging. To address this, we applied THIR across all magnifications of the data set using the most commonly used values of $K$. Following the evaluation protocol of [32, 51, 52], we conducted a fair comparison with several state-of-the-art methods.

At 400× magnification, THIR achieves an accuracy of 0.98 at the top-5, outperforming both supervised and unsupervised methods. Notably, it delivers an accuracy of 0.98, which is 18% higher than Breasttwins (0.69), a fully supervised method utilizing a Siamese network for similarity learning. In [14], FedCBMIR was designed to generalize across all magnifications and was evaluated at 400×. Although FedCBMIR aims to enhance performance on unseen data, THIR consistently delivers better retrieval accuracy and precision, even without any learning phase or hyperparameter tuning. This trend continues at 200× magnification with $K = 5$, where THIR achieves a precision of 0.99, surpassing FedCBMIR by 10% (FedCBMIR precision = 0.89). Additionally, THIR demonstrates stronger performance than other baselines like CNN-based AE (0.93 precision) and MCCH (0.89 precision).

In Table 2, we include three methods from [53], for which the value of $K$ used in top-$K$ retrieval is not explicitly defined in the original paper. The only instance where a specific $K$ value is mentioned appears in the caption of a qualitative figure, where they state $K = 5$. However, this information is not provided for the quantitative results reported in the tables. Therefore, due to the lack of clarity regarding the retrieval threshold, we marked the $K$ value as "Not-defined" in our comparison table. The paper [53] focuses on hashing-based methods and reports retrieval performance at various code lengths: 16, 32, and 64 bits. To include these results in our comparison, we selected the highest performance across all bit lengths. For example, the DCMMH

method achieves its best performance (0.95) at 32 bits for 40× magnification, while DPSH reaches its highest at 16 bits for the same magnification. To ensure a fair comparison with our proposed method, we report only the top-performing results from each approach, regardless of the bit length used.

Across all magnifications and for both $K = 3$, $K = 5$, and $K = Not\ defined$, THIR maintains high and stable performance. It consistently outperforms CNN-based AE, MCCH, and several hashing-based methods (e.g., HashNet, IDHN, DTQ), as well as state-of-the-art frameworks such as FedCBMIR and VTHC. These results highlight the robustness and effectiveness of TDA in CBMIR, especially compared to DL-based methods that require substantial training time and parameter optimization.

Figure 4 shows four random examples of retrieval results with corresponding labels. Incorrect retrievals (where the retrieved label differs from the query label) are outlined in red. This visual analysis enables qualitative evaluation of system performance and highlights the capability of topological features to capture structural similarities in histopathological breast cancer images. Beyond label checking, pathologists can also examine structural patterns between queries and retrieved results based on their expertise.

Figure 5 contains four panels, each showing the Betti curves of four randomly selected images. Different colored lines represent different images. A guide bar on top of each panel indicates the corresponding class label of each image related to its curve. The x-axis displays the filtration steps, while the y-axis shows the Betti values. The x-axis ranges from 0 to 600, illustrating that 600 features were extracted from the Betti values of each RGB image. As mentioned earlier, we computed the Betti values for each channel separately using the cubical complex and

then concatenated them to obtain a representative feature vector for the entire image. Thus, the x-axis can be interpreted in intervals of 200: the interval [0–200] represents the Betti values for the red channel, [200–400] for green, and [400–600] for blue. This figure demonstrates how the trend of Betti curves behaves across channels for images from the same or different classes. For instance, in the top-left panel, all images belong to the same class (Benign), and their Betti curves follow a similar trend, indicating topological similarity. In contrast, in the top-right panel, the red line deviates noticeably, suggesting a different topological pattern compared to the other three images. The remaining three (blue, orange, and green lines) follow a similar trend, which reflects their similarity in class labels. In the bottom panels, two additional examples show that two images have similar Betti curves and share the same labels, while the other two follow distinct trends and have different labels. These characteristic patterns suggest that Betti curves encode discriminative information suitable for unsupervised CBMIR.

Furthermore, Figure 6 demonstrates the retrieval of four random queries based on Betti values. In each retrieval panel, four Betti curves are shown: the red line represents the query, while blue, orange, and green lines represent the top-3 retrieved images. The class labels are indicated above each image, and a guide bar links the curve colors to the images. In some cases, such as the top-left panel, the retrieved image shares similar topological features with the query but has a different label, suggesting intraobserver variability. In other cases, such as the top-right panel, all retrieved images share the same label with the query, reinforcing the robustness of Betti features. Such visualization suggests that images with similar Betti curves may share underlying histopathological characteristics, even when labeled differently. This highlights the potential of CBMIR for digital pathology applications and demonstrates advantages over traditional Computer-Aided Diagnosis (CAD) tools.

As an indirect comparison between the THIR result and state-of-the-art classifiers on the same images, Table 3 provides comprehensive information regarding the accuracy of classifiers on the BreaKHis data set at $40X$. TopOC-1 and TopOC-CNN in [45] obtained 89% and 93% accuracy, while THIR was successful in retrieving images with the same label with 98% accuracy. However, TopOC-1 and TopOC-CNN needed some training time and hyperparameter tuning.

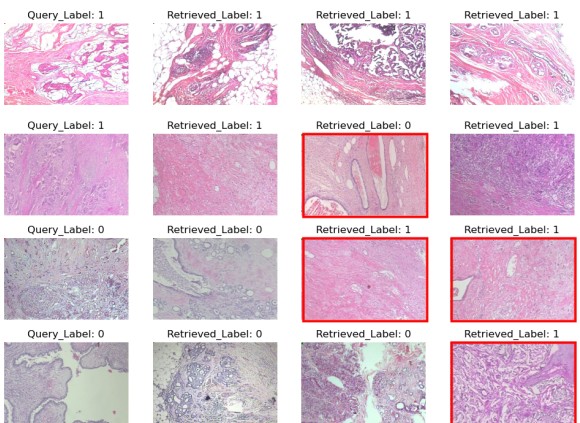

**Figure 4.** shows four random queries and their similar patches. Each row represents a query image (leftmost) followed by its top-3 retrieved images. The true class labels are shown as Query Label and Retrieved Label. Misclassified retrievals are outlined in red.

## 5.2 How Fast, How Simple: Training, Searching, and Using the Model

Several recent CBMIR studies based on CAE have reported training times of approximately 9.33 hours

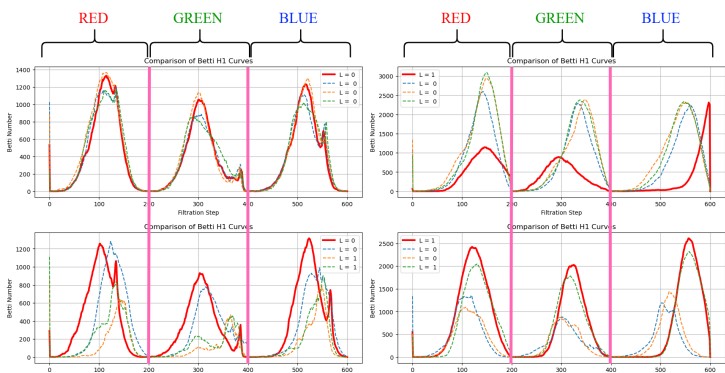

**Figure 5.** shows four panels, each with four concatenated Betti curves for four random images. The label of each image is represented in a small bar on top of each panel. $L = 0$ and $L = 1$ mean "Benign and "Malignant" cases, respectively. This explains how images with the same cancer grade have similar Betti curves. The y-axis illustrates the number of loops at each filtration step, and the x-axis represents the filtration step for each channel. $R = 200$ yields 200 filter steps for each channel, which means 600 filter steps in total for an RGB image.

**Table 2.** Performance comparison of THIR with recent methods at various $K$ values on BreaKHis across all magnifications. The best results per metric are bolded.

| Magnification | Method | $K$ | Accuracy | Recall | Precision | F1-score |
|---|---|---|---|---|---|---|
| | **THIR** | 3 | 0.95 | **0.97** | 0.96 | **0.97** |
| | **THIR** | 5 | **0.98** | **0.97** | 0.96 | 0.95 |
| | FedCBMIR [14] | | 0.97 | - | 0.96 | **0.98** |
| | CBMIR [14] | 5 | 0.95 | - | 0.93 | 0.96 |
| | MCCH [54] | | - | - | 0.94 | - |
| | CNN-based AE [52] | | - | 0.77 | 0.95 | - |
| | HSDH [55] | | - | - | **0.99** | - |
| 40× | DTQ [55] | | - | - | 0.91 | - |
| | ATH [55] | 400 | - | - | 0.89 | - |
| | IDHN [55] | | - | - | 0.95 | - |
| | HashNet [55] | | - | - | 0.91 | - |
| | VTHC [56] | 6 | - | - | 0.98 | - |
| | DCMMH [53] | | - | - | 0.96 | - |
| | DPSH [53] | Not defined | - | - | 0.95 | - |
| | ADSH [53] | | - | - | 0.95 | - |
| | **THIR** | 3 | 0.97 | 0.98 | 0.98 | 0.98 |
| | **THIR** | 5 | **0.99** | **0.99** | **0.99** | **0.99** |
| | FedCBMIR [14] | | 0.94 | - | 0.92 | 0.96 |
| | CBMIR [14] | 5 | 0.90 | - | 0.88 | 0.94 |
| 100× | MCCH [54] | | - | - | 0.92 | - |
| | CNN-based AE [52] | | - | 0.49 | 0.77 | - |
| | VTHC [56] | 6 | - | - | 0.99 | - |
| | DCMMH [53] | | - | - | 0.95 | - |
| | DPSH [53] | Not defined | - | - | 0.95 | - |
| | ADSH [53] | | - | - | 0.94 | - |
| | **THIR** | 3 | 0.97 | 0.98 | 0.98 | 0.98 |
| | **THIR** | 5 | **0.99** | **0.99** | **0.99** | **0.99** |
| | FedCBMIR [14] | | 0.92 | - | 0.89 | 0.94 |
| | CBMIR [14] | 5 | 0.89 | - | 0.87 | 0.93 |
| 200× | MCCH [54] | | - | - | 0.91 | - |
| | CNN-based AE [52] | | - | 0.76 | 0.92 | - |
| | VTHC [56] | 6 | - | - | 0.98 | - |
| | DCMMH [53] | | - | - | 0.97 | - |
| | DPSH [53] | Not defined | - | - | 0.96 | - |
| | ADSH [53] | | - | - | 0.95 | - |
| | **THIR** | 3 | 0.95 | 0.97 | 0.95 | 0.97 |
| | **THIR** | 5 | **0.98** | **0.99** | **0.98** | **0.99** |
| | FedCBMIR [14] | | 0.96 | - | 0.94 | 0.97 |
| | UCBMIR [30] | | - | 0.79 | 0.91 | - |
| 400× | Breast-twins [54] | 5 | 0.69 | 0.82 | 0.91 | 0.81 |
| | MCCH [31] | | - | - | 0.89 | - |
| | CNN-based AE [52] | | - | 0.69 | 0.93 | - |
| | VTHC [56] | 6 | - | - | 0.99 | - |
| | DCMMH [53] | | - | - | 0.96 | - |
| | DPSH [53] | Not defined | - | - | 0.95 | - |
| | ADSH [53] | | - | - | 0.95 | - |

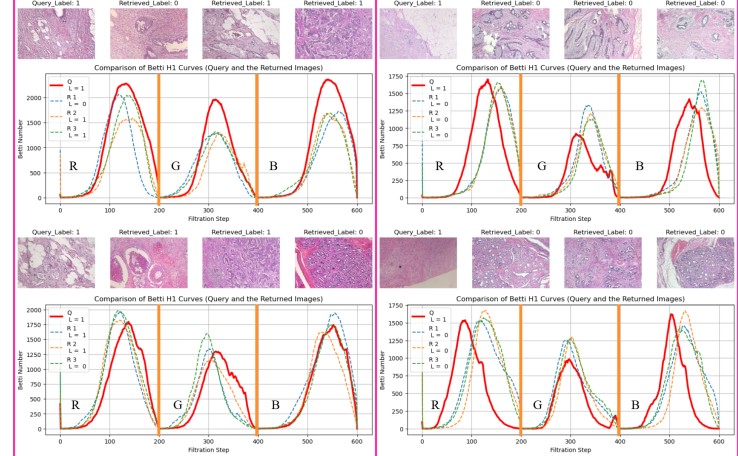

**Figure 6.** demonstrates the retrieval of four random queries using Betti values. There are four panels of Betti curves with the corresponding queries and returned images. For each panel of Betti curves, a query image (top left) is compared to a set of retrieved images using the Euclidean distance function. Each curve in the panel reflects topological features over the filtration steps. The alignment of Betti curve patterns with the query supports the effectiveness of PH for topology-aware image retrieval. The label of each image is represented in a small bar on top of each panel. $L = 0$ and $L = 1$ mean "*Benign* and "*Malignant*" images, respectively.

and 6.59 hours on the BreaKHis dataset [14] at 40× magnification, using an NVIDIA Tesla T4 GPU. Although FedCBMIR reduces training time compared to traditional CBMIR, it still requires a substantial training duration. In DL-based methods, once the models are trained, features are subsequently extracted for CBMIR tasks; however, the feature extraction time is often not reported in these studies. In contrast, our method bypasses the lengthy training phase entirely. THIR efficiently processes the entire dataset by performing the filtration, calculating the Betti values $\beta_1$, and extracting 600 features per image. This comprehensive feature extraction for all images in the dataset requires merely about 20 minutes, which is significantly faster than the combined training and feature extraction time of existing deep learning approaches.

**Table 3.** Classification accuracy of THIR compared to state-of-the-art methods on the BreaKHis dataset at 400× magnification, with $K = 5$ for THIR.

| Method | Accuracy |
|---|---|
| **THIR** | **0.98** |
| **DenseNet201** [57] | 0.95 |
| **IDSNet** [58] | 0.94 |
| **VGG16** [59] | 0.94 |
| **Resnet** [60] | 0.94 |
| **DenseNet201** [61] | 0.89 |
| **BkCapsNet** [30] | 0.88 |
| **CapsNet** [62] | 0.88 |
| **BkNet** [63] | 0.84 |
| **MobileNet** [64] | 0.84 |
| **AlexNet** [65] | 0.81 |

# 6 Conclusion and future work

This paper introduces THIR, an unsupervised, training-free, and interpretable framework for Content-Based Medical Image Retrieval (CBMIR) based on topological data analysis. Using cubical persistence and Betti values for loops ($\beta_1$), THIR extracts robust topological fingerprints from raw RGB histopathological images without annotations, GPU acceleration, or hyperparameter tuning.

Experiments on the BreaKHis dataset show that THIR consistently outperforms supervised and unsupervised baselines across all magnifications. At 400×, it improves accuracy by 31% and precision by 18% over Breast-twins, a Siamese-network-based model. At 200×, it achieves a 10% gain in precision over FedCBMIR. Similar margins are observed at lower magnifications, with precision up to 0.99 at 100×, surpassing all compared methods. These results confirm THIR as a strong alternative for CBMIR in label-scarce or resource-constrained settings.

THIR extracts 600 topological features per image in under 20 minutes on a standard CPU, offering an efficient and scalable solution compared to deep learning models requiring extensive training. Its performance, efficiency, and interpretability make it suitable for clinical applications and diagnostic support.

Future work will explore higher-dimensional features (e.g., persistence images, landscapes), extension to multi-channel and multi-organ datasets, alternative color spaces beyond RGB, and generalization to multi-class and whole-slide image retrieval tasks.

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
