# OpenReview forum: "THIR: Topological Histopathological Image Retrieval"
_NLDL.org/2026/Conference — Submitted to NLDL 2026_

### Official Review · Reviewer_h2ZS · 2025-09-25
**Novel and efficient approach to CB-MIR**

**Rating:** 4
**Confidence:** 4
**Final Rating:** 4
**Final Confidence:** 4

**Summary:**

This paper presents THIR, a CB-MIR approach leveraging Topological Data Analysis to extract multi-resolution topological, with a focus on the number of loops ($β_1$). Unlike deep learning-based descriptors, which dominate the current state of the art, the proposed method is unsupervised and does not require high-performance computing resources. The authors claim that THIR outperforms both supervised and unsupervised CB-MIR methods on several benchmarks, positioning it as a strong alternative to existing approaches.

**Strengths:**

- The paper provides a thorough literature review, situating the work well within the context of CB-MIR and TDA.
- The experimental design is sound, with appropriate metrics and a broad selection of related approaches for comparison.
- The figures comparing per-channel Betti values provide valuable support for the claim of interpretability, illustrating how topological features - relate to image structure.
- Despite some brittleness in the experimental execution, the results generally support the good performance and convenience of the proposed descriptor.

**Weaknesses:**

- The proposed method is essentially a feature descriptor combined with Euclidean distance, rather than a full retrieval framework, as the proposed ranking strategy and similarity/distance metric are the default ones. This does not reduce the merit of the paper at all. I just consider the authors should be more honest at presenting as a novel descriptor that was tested with |Euclidean distance.
- Evaluation is limited to a single dataset, which restricts the ability to recommend the descriptor for general use in the histopathology domain. By checking the results, looks like BrakHis is saturated in the sense that the state of the art is already close to a 99% performance, so any improvement is infimal. I recommend the authors mentioning this fact and using this as an opportunity to higlight the convenience of using a descriptor that does not require training or GPUs.
 - The use of different values of K across methods hinders proper comparison. While the authors acknowledge this, I recommend presenting an aditional table with filtered results where only appropriately comparable methods are displayed (e.g., all the ones that reported K=3 together to spot more easily how THIR performs).
- It is unclear whether statistical significance tests were performed. I know it is difficult to align such a thing accross multiple papers, but the authors must be clear about the fact that they could (or couldn't) perform multiple runs, or maybe cross validation, or if the comparion is based only on single runs.
- The description of general CB-MIR systems could be shortened in favor of a more detailed explanation of TDA and persistence diagrams.

**Final Justification:**

Thanks to the author for their answer to my concerns. I'm happy to keep my decision of acceptance of the paper.

**Justification:**

While there are some concerns regarding the presentation and depth of methodological detail and the generalization capacity of the descriptor, the paper introduces a novel and efficient approach to CB-MIR using topological features. The evaluation appears generally correct, and the method demonstrates competitive performance without the need for supervision or specialized hardware. I recommend acceptance, as the work offers valuable insights and practical contributions to the field.

---

> ### Author Rebuttal · Authors · 2025-10-22
>
> Q1. The proposed method is essentially a feature descriptor combined with Euclidean distance, rather than a full retrieval framework, as the proposed ranking strategy and similarity/distance metric are the default ones. This does not reduce the merit of the paper at all. I just consider the authors should be more honest at presenting as a novel descriptor that was tested with |Euclidean distance.
> - Regarding the CBMIR framework, to ensure a fair comparison, we kept all other components consistent with previous studies and followed the same overall pipeline, allowing us to focus solely on the feature descriptors. We will add a line to the paper to clarify this point. Thank you for reminding us of this very important and noteworthy detail.
>
> Q2. Evaluation is limited to a single dataset, which restricts the ability to recommend the descriptor for general use in the histopathology domain. By checking the results, looks like BrakHis is saturated in the sense that the state of the art is already close to a 99% performance, so any improvement is infimal. I recommend the authors mentioning this fact and using this as an opportunity to higlight the convenience of using a descriptor that does not require training or GPUs.
> - We greatly appreciate your time and your very insightful comments. We agree that research on the BreaKHis dataset is almost saturated, and this was precisely our reason for choosing it. As one of the most commonly used datasets in the CBMIR domain, it allows for broad and meaningful comparisons. Our aim was to demonstrate that an unsupervised TDA-based approach can achieve comparable or even higher accuracy than DL/ML methods, without requiring large training sets or labeled data. By evaluating BreaKHis across all magnifications, we were able to compare our method with a wide range of existing approaches. Thank you for your recommendation. We will update the paper in the camera-ready version to better emphasize the advantages of using a descriptor that does not require training or GPU resources.
>
> Q3. The use of different values of K across methods hinders proper comparison. While the authors acknowledge this, I recommend presenting an additional table with filtered results where only appropriately comparable methods are displayed (e.g., all the ones that reported K=3 together to spot more easily how THIR performs).
> - The reason we included a long table is that, although BreaKHis is a common dataset in this domain, the K value varies considerably across different studies. Since the main goal of this paper is to demonstrate that TDA-based features can perform comparably to DL/ML methods, it was important for us to account for all such variations, including the choice of K. To address this concern, we will update the paper to include an additional concise table summarizing the results for K = 3.
>
> Q4. It is unclear whether statistical significance tests were conducted. I understand it is challenging to standardize this across multiple papers, but the authors must clarify whether they performed multiple runs, used cross-validation, or if the comparison was based solely on single runs.
> - All experiments were performed using fixed data split across each magnification and method to ensure comparability. Future work will incorporate multiple randomized splits and significance analysis to assess robustness.
>
> Q5. The description of general CB-MIR systems could be shortened in favor of a more detailed explanation of TDA and persistence diagrams.
> - In the updated version of the paper, we will add some more details regarding TDA and reduce some parts of the CBMIR explanation.
>
> Thank you for taking the time to review our paper and help improve the manuscript.

---

### Official Review · Reviewer_Dga7 · 2025-10-07
**Novel framework to  retrieve histopathological images based on their intrinsic structural patterns. Is a good paper.**

**Rating:** 4
**Confidence:** 3

**Summary:**

The paper proposes a novel framework, THIR. This is a training-free, fully unsupervised content-based medical image retrieval (CBMIR) framework for histopathology based on TDA. The method computes cubical persistent homology separately on the RGB channels of each image, composes Betti curves at a chosen resolution, concatenates the three per-channel curves into a 600-dimensional descriptor, and retrieves top-$K$ neighbors using Euclidean distance.
Claims emphasize interpretability and efficiency versus DL baselines and hashing methods.

**Strengths:**

The paper’s key strengths lie in its simplicity, efficiency, and interpretability. THIR is a training-free, fully unsupervised CBMIR pipeline that avoids hyperparameter sweeps and runs end-to-end on CPU in a short time, making it an accessible and reproducible baseline. By building descriptors from Betti curves derived via cubical persistence on each RGB channel, the method provides a transparent shape/texture representation that aligns with clinical explainability goals.

**Weaknesses:**

The main limitations concern evaluation breadth, comparison rigor, and analysis depth. Results are confined to a single, binary dataset  without external validation on other organs, stains, or multi-class/WSI-scale scenarios, and there are no user studies to support clinical applicability.

**Justification:**

THIR is a neat, interpretable, and surprisingly competitive baseline for histopathology retrieval that avoids training and GPUs. Its clarity and practical value are notable, and the qualitative visualizations are compelling. However, the study would be substantially stronger with rigorous, standardized retrieval protocols across baselines.

---

> ### Author Rebuttal · Authors · 2025-10-22
>
> Q. The main limitations concern evaluation breadth, comparison rigor, and analysis depth. Results are confined to a single, binary dataset without external validation on other organs, stains, or multi-class/WSI-scale scenarios, and there are no user studies to support clinical applicability.
> - Thanks for your insightful comments. We completely understand your concerns regarding the evaluation on other datasets or multi-class datasets.
> BreaKHis was selected as our dataset to first analyze and evaluate the performance of Betti-1 values in the CBMIR domain. This choice was made after reviewing recent studies, as most existing CBMIR works and methods have been evaluated on this dataset, making it the most common benchmark in this field. Using BreaKHis therefore allows for a fair and diverse comparison with previous approaches. This serves as a foundation for further analysis of Betti values in multi-class datasets.
> For other multi-class datasets such as SICAPv2, PANDA, Cam17, and Cam16, only a limited number of studies are available, which would not allow us to provide the same breadth of comparison between DL/ML-based methods and our proposed TDA-based approach. The main focus of this paper is to highlight the advantages of using a topological descriptor that does not require training or GPU resources, in contrast to heavy training-based processes. The method achieves competitive performance without the need for supervision or specialized hardware.
> Thanks to your comment, we will update the manuscript to clarify the reason that we selected this data set.
> Additionally, as mentioned in the Conclusion and Future Work section, this framework can be further extended to multi-organ studies in the future.
>
>
> Thank you for taking the time to review our paper and help improve the manuscript.

---

### Official Review · Reviewer_2fjP · 2025-10-09
**Interesting study but with weaknesses regarding the experimental section leaving applicability in clinical settings questionable**

**Rating:** 2
**Confidence:** 3
**Final Rating:** 4
**Final Confidence:** 5

**Summary:**

The authors propose using TDA features for a CBIR of H&E stained tissue images. They are fast to compute and unlike dense CNN feature extractors invariant to rotations. However the experiments were only conducted on a single data set of breast tissue containing images of malignant and benign tissue. This is a very simplified scenario compared to using such a system in real clinical settings when one would like to query rare conditions, or more specific tissue regions, and that across hospitals and labs where variations in staining and scanning can be expected, hence it remains unexplored how useful the proposed retrieval system would be.

**Strengths:**

- Good literature and background section
- Manuscript is well-written and easy to read
- Nice idea to use TDA in CBIRs
- The feature extraction to index the CBIR database is efficient and computationally cheap
- The method was compared to a number of competing methods

**Weaknesses:**

The main weakness is the limited evaluation of testing the method only on a single breast cancer dataset with binary labels (malignant/healthy) which is a classification task that is relatively easy to do visually for a human for this particular cancer type, hence we can only draw limited conclusions of how well such a setup would work on more challenging tissue like WSI of prostate biopsies, nor is it likely that in its current form, or more fine-grained tissue categorizations (e.g. molecular subtypes of breast cancer etc). The information provided in the manuscript on the dataset and the evaluation of the experiments also lacks highly relevant details which I urge the authors to address, see below:

- Details on the dataset are very vague. Are these whole slide images, tissue micro array cores, patches of the WSIs? What resolution and field of view do these images have? Please add this information, otherwise it is completely unclear for what type of data this method has been evaluated on.
- How many images of your dataset do you use to create the indexed database, how many do you use as a test set of query images for evaluation of the CBIR? Are the query images independent from the database, i.e. none of the database and query images are extracted from the same WSI or same patient? Add these essential details in the data description.
- Only one dataset is used, so it is not clear how this method generalizes
- I do not understand the sentence “Since the images are from the cancerous tissue, they are not affected by image transforming, inverting, zooming in, or rotation by 90 degrees”. First, what is meant by “image transforming”, that could be anything? And clearly some transformations will destroy the relevant information that is needed for the downstream task, so specify what transformations you mean. Second, what do you mean by “since the images are from cancerous tissue, they are not affected”. Why is this specific to cancerous tissue and not any tissue in general?
- Clarify what you consider a retrieval success - I assume it is to retrieve another benign image if the query is benign, or a malignant one if the query is malignant? Should they be of the same magnification?
- You introduce the variable th, use it also in Fig 2 for consistency
- As the Betti curves are computed on the RGB images, it is likely that this will not generalize well to different staining procedures when images were taken at different hospitals following different staining procedures or using different WSI scanners. I would suggest to consider deconvolving the RGB channels into an H and an E stain channel (then each channel and associated persistence diagram captures different structures that are stained by H or E respectively), and/or learn a persistence diagram that is invariant to the variations encountered in staining in large datasets coming from multiple sources (which is ultimately where a CBIR is useful, e.g. by being able to fetch images of rare cancer types from other hospitals that are similar to a query I have).
- Clarify what you mean with R resolution in line 368 and following
- Lines 408, 409: incomplete sentence
- Clarify your evaluation measures. For each query image, you retrieve k images. What does the accuracy you report mean in this case? Do you compute how often you have one match of the same label among the top k retrieved images, or do you compute how many in the top-k are of the same label and aggregate that over your set of query images?


minor remarks: “”In this binary data set, tissues were stained with Hematoxylin
and Eosin (H&E), which is the most common color
in histopathological images”,: H&E is not a color but a tissue stain that gives the tissue color

**Final Justification:**

The study has a number of limitations and I do not believe that it is generalizable to other data in its current state, hence the impact and applicability for use cases is limited. However, the authors were able to give arguments for the contributions in the author rebuttal, which I hope they will include to discuss the limitations in a final version of the submission, and make clear that this is an initial study aiming to investigate the potential of topological descriptors in CbIR systems, rather than suggesting they have reached a validated solution.

**Justification:**

Details about the data and evaluation are lacking which makes it hard to draw conclusions from the results. I believe however that this can be improved easily and I do absolutely like the idea of the study and think it has a lot of potential.

---

> ### Author Rebuttal · Authors · 2025-10-22
>
> Q1. The main weakness is the limited evaluation of testing the method only on a single breast cancer dataset with binary labels (malignant/healthy) which is a classification task that is relatively easy to do visually for a human for this particular cancer type, hence we can only draw limited conclusions of how well such a setup would work on more challenging tissue like WSI of prostate biopsies, nor is it likely that in its current form, or more fine-grained tissue categorizations (e.g. molecular subtypes of breast cancer etc).
> - We completely understand your concerns and point of view regarding the dataset. Here, we would like to emphasize that our main goal was to demonstrate the performance of our method in comparison with DL/ML-based approaches, which require a wide diversity of datasets to ensure a fair comparison. Although other datasets such as SICAPv2, PANDA, Cam17, and Cam16 exist, they are less frequently used in the CBMIR context and do not provide enough benchmarks to clearly highlight the advantages of using a topological descriptor, which requires neither training nor GPU resources, over computationally intensive training processes. Our method achieves competitive performance without supervision or specialized hardware.
> By reviewing recent studies in the CBMIR domain, we found that BreaKHis is the most widely adopted dataset that fulfills our need to enable a full and fair comparison with existing DL/ML-based methods.
> It is also worth noting that the BreaKHis dataset contains both benign and malignant tumors, which can be categorized into different types based on their microscopic appearance. These subtypes have different prognoses and potential treatment implications. Specifically, the dataset includes four histologically distinct types of benign breast tumors, adenosis (A), fibroadenoma (F), phyllodes tumor (PT), and tubular adenoma (TA), and four malignant tumors, ductal carcinoma (DC), lobular carcinoma (LC), mucinous carcinoma (MC), and papillary carcinoma (PC). However, most prior studies have approached this dataset as a binary classification problem (benign vs. malignant).
> Although our work focuses on the binary classification setting (benign vs. malignant), it is important to acknowledge that the presence of these eight subtypes inherently affects retrieval behavior. Therefore, the dataset cannot be regarded as a simple binary collection, as these intra-class variations play a significant role in the overall retrieval performance.
> In the updated version of the paper, we will include additional details to clarify the rationale and objectives behind selecting this dataset.
>
> Q2. The information provided in the manuscript on the dataset and the evaluation of the experiments also lacks highly relevant details which I urge the authors to address, see below:
> Details on the dataset are very vague. Are these whole slide images, tissue micro array cores, patches of the WSIs? What resolution and field of view do these images have? Please add this information, otherwise it is completely unclear for what type of data this method has been evaluated on.
> - In this paper, we followed the same structure as other papers to divide the data into the test and train sets to have a fair comparison.
> Based on your comments and questions regarding the data set, here, we would like to add some more details about it.
> The BreaKHis dataset does not contain whole-slide images (WSIs) or tissue microarray (TMA) cores.
> Instead, it consists of microscopic image patches of breast tumor tissue obtained from biopsy slides.
> Each image corresponds to a specific magnification level acquired using a microscope coupled to a digital camera. Each image represents a local field of view of approximately 0.25 mm² to 0.04 mm², depending on the magnification.
> Your very smart concerns regarding the data set help us to update the manuscript with more details about it in the next version of the paper.
>
> Q3. How many images of your dataset do you use to create the indexed database, how many do you use as a test set of query images for evaluation of the CBIR? Are the query images independent from the database, i.e. none of the database and query images are extracted from the same WSI or same patient? Add these essential details in the data description.
> - We appreciate your question regarding the independence of images in the BreaKHis dataset.
> The BreaKHis dataset is composed of microscopic image patches acquired at four magnifications (40×, 100×, 200×, 400×) from biopsy slides of 82 patients, distributed across eight histopathological subclasses (four benign and four malignant).
> Each patient contributes multiple fields of view, which may correspond to different tissue regions and magnifications.
> Consequently, images originating from the same patient can exhibit marked morphological variability due to the intrinsic heterogeneity of breast tumors. Even within a single tissue section, pathologists often observe different local grades and structures, reflecting variations in gland formation, nuclear atypia, and stromal composition.
> Therefore, our CBIR evaluation focuses on image-level (patch-level) retrieval, where each image represents a distinct microscopic field rather than a single patient identity.
> This design is consistent with how BreaKHis has been used in prior CBIR and classification studies, which also assess models at the patch level. In the scope of this paper, we mainly focused on retrieving similar patches at the same magnification.
> We will clarify this point in the revised manuscript to emphasize that our evaluation targets the system’s ability to retrieve histologically similar patterns, not patient-specific characteristics.
>
> Q4. Only one dataset is used, so it is not clear how this method generalizes.
> - It is explained above.
>
> Q5. I do not understand the sentence “Since the images are from the cancerous tissue, they are not affected by image transforming, inverting, zooming in, or rotation by 90 degrees”...
> - We appreciate the reviewer’s careful reading and valuable feedback. We agree that the phrasing could be clearer and more specific. In the revised version, we will refine this statement to be a better description.
>
> Q6. Clarify what you consider a retrieval success ...
> - According to (Hegde et al., “SMILY: Similar Image Search for Histopathology,” NPJ Digital Medicine, 2019) a retrieval result is considered correct if the returned images share the same label as the ground truth. In Figure 6, we illustrated the potential for intra-observer variability among pathologists. In response to the reviewer’s concerns regarding the evaluation, we will further clarify the definition of a correct retrieval based on the recent papers already cited in the evaluation section. In the current version of the paper, we mention that we follow the same evaluation methodology as these prior works. In the updated version, we will include additional details to make this evaluation process clearer.
>
> Q7. You introduce the variable th, use it also in Fig 2 for consistency. Lines 408, 409: incomplete sentence. minor remarks: “”In this binary data set, tissues were stained with Hematoxylin and Eosin (H&E), Clarify what you mean with R resolution...
> - Thanks for your very complete and detailed comments. In the case of some modifications in the paper and the figures, such as Line 408,409, R in Line 368, Figure 2, and the H&E staining, we will modify the paper in its updated version.
>
> Q8. As the Betti curves are computed on the RGB images, it is likely that this will not generalize well to different staining procedures when images were taken at different hospitals following different staining procedures or using different WSI scanners...
> - The current paper uses RGB channels because this is the dataset’s native color space, and we have already discussed the importance of color information in digital pathology. We also indicate that exploring alternative color spaces is part of our future work. Additional details will be added to the limitations section.
> It is also worth mentioning that by applying a cubical complex, we do not rely on a single thresholding value such as Otsu’s method. Instead, we perform multiple thresholding steps over each unique pixel intensity, effectively covering most intensity levels across the image. This approach helps mitigate the impact of color variation to some extent.
> Thanks to your point, we will update the paper by adding extra information in this regard.
>
>
>
> Thank you for taking the time to review our paper and help improve the manuscript.

---

### Official Review · Reviewer_tkRx · 2025-10-09
**Topological Histopathological Image Retrieval**

**Rating:** 4
**Confidence:** 4

**Summary:**

The article presents a topological data analysis approach for generating a topological signature for images, and demonstrates a use case for content based image retrieval for histopathology images of breast cancer tissue. The experimental part presents a comparison of the proposed topological histopathological image retrieval (THIR) method to deep learning based methods presented in the literature. Although the results demonstrate high level of performance for THIR and outstanding performance compared to deep learning based methods, it is not fully clear whether the experimental setup is fully comparable. The article is a sound and concise study, albeit may be slightly out of the focus of the DL conference.

**Strengths:**

THIR is presented as a simple and efficient feature extraction method, converting an image into a feature vector of 600 elements. Experimental results are convincing, although division into train/test for the dataset should be clarified.

**Weaknesses:**

Experimental results appear to be comparisons of the results obtained with the proposed THIR framework against results presented in the literature. It should be clarified whether the results are obtained using similar division to enable direct comparison.
Discussion and examples of unsuccessful classification/retrieval results is completely missing.
Abstract begins with selling the work using the cancer as a context - this is overselling as the study is very much a technical one without any direct implication on any practical clinical value.
Topological data analysis is rather a traditional image processing approach than a DL approach - is this study in the focus of the event?

**Justification:**

Study presents a fresh non-DL approach for content based image retrieval using breast cancer histopathology images as a case study. The method is interesting and contains novelty.

---

> ### Author Rebuttal · Authors · 2025-10-22
>
> Q1. Experimental results appear to be comparisons of the results obtained with the proposed THIR framework against results presented in the literature. It should be clarified whether the results are obtained using similar division to enable direct comparison.
> - Thanks for your insightful comments. That is a very good point that was missing in the paper. I will update the paper with information regarding the data division in the next phase of the conference so that we can revise the manuscript accordingly.
> In this paper, to ensure a fair comparison, I used the same Train/Test division of the data. Although there is no explicit training phase in this study, the same training and test sets as in previous studies were considered for consistency between the previous cases (database) and the queries.
> Q2. Discussion and examples of unsuccessful classification/retrieval results are completely missing.
> - We already presented some mismatches in the paper. For instance, Figure 4 marks incorrect top-K retrievals with red outlines, which serve as concrete examples of failures (e.g., benign queries retrieving malignant patches or vice versa). Figure 6 (top-left panel) also shows a case where the retrieved image’s label differs from the query despite a close Betti-curve match; in the text, we note that this may reflect intra-observer variability or overlapping morphology. In the updated version of the paper, we will include additional details to make these failure cases clearer.
>
> Q3. Abstract begins with selling the work using the cancer as a context - this is overselling as the study is very much a technical one without any direct implication on any practical clinical value.
> - We completely understand your point regarding the abstract. Our intention in starting the abstract that way was to highlight the potential importance of this tool in the context of breast cancer. We will revise the abstract in the camera-ready version to better reflect the technical focus of the study.
> We will update the abstract with a new format as below:
> Histopathological image analysis plays a central role in supporting cancer diagnosis and research. In this study, we present THIR, a Content-Based Medical Image Retrieval (CBMIR) framework that applies Topological Data Analysis (TDA), specifically, Betti numbers derived from persistent homology, to characterize and retrieve histopathological images based on their intrinsic structural patterns. Unlike deep learning approaches that require extensive training, annotated datasets, and high computational resources, THIR operates in a fully unsupervised and training-free manner. It extracts topological fingerprints directly from RGB histopathological images using cubical persistence, encoding the evolution of loops as compact and interpretable feature vectors. Similarity retrieval is then performed by computing distances between these topological descriptors, efficiently returning the top-K most relevant matches.
> Experiments on the BreaKHis dataset demonstrate that THIR achieves competitive or superior retrieval performance compared to several state-of-the-art supervised and unsupervised methods. The framework processes the entire dataset in under 20 minutes on a standard CPU, providing a fast, scalable, and interpretable approach for histopathological image retrieval.
>
> Q4. Topological data analysis is rather a traditional image processing approach than a DL approach - is this study in the focus of the event?
> - About your concern related to the scope of the conference, although our method is not a DL-based method, we contribute a training-free yet competitive approach that outperforms or matches DL baselines on BreaKHis across magnifications (Table 2 and Section 5.1). NLDL welcomes methods that advance learning-centric tasks (in our case, similarity learning/retrieval) even if the representation is training-free. THIR removes supervision and training while improving retrieval accuracy vs. state-of-the-art DL pipelines, which are directly relevant to robust, data-efficient medical AI.
>
>
> Thank you for taking the time to review our paper and help improve the manuscript.

---

### Meta-Review · Area_Chair_ZF3B · 2025-11-03

**Recommendation:** Reject
**Confidence:** 5

**Metareview:**

dear authors,

thanks for your submission to NLDL 2026. While your paper definitely bring some value and a fresh view on an important problem, I do not consider it falls in the scope of the NLDL conference, that aims to present new and original research on all aspects of Deep Learning.

I hope the reviews will help you to improve your work and I strongly recommend to consider a more relevant venue, e.g. medical imaging (MICCAI) or image retrieval (CBMI).

---

### Decision · Program_Chairs · 2025-11-05

**Decision:**

Reject

**Comment:**

After discussing with the AC, the PCs agree that the paper is high quality but outside the conference scope. We see its potential and encourage submission to a more appropriate venue.